# Learning to Defer with an Uncertain Rejector via Conformal Prediction

**Yizirui Fang**
Department of Computer Science
Johns Hopkins University
yfang52@jhu.edu

**Eric Nalisnick**
Department of Computer Science
Johns Hopkins University
nalisnick@jhu.edu

## Abstract

*Learning to defer* (L2D) allows prediction tasks to be allocated to a human or machine decision maker, thus getting the best of both's abilities. Yet this allocation decision depends on a 'rejector' function, which could be poorly fit or otherwise misspecified. In this work, we perform uncertainty quantification for the rejector sub-component of the L2D framework. We use conformal prediction to allow the reject to output sets, instead of just the binary outcome of 'defer' or not. On tasks ranging from object to hate speech detection, we demonstrate that the uncertainty in the rejector translates to safer decisions via two forms of selective prediction.

## 1   INTRODUCTION

*Learning-to-Defer* (L2D) by [11, 14] is a framework for human-AI collaboration that divides responsibility between machine and human decision makers. For every test instance, a 'rejector' function decides if the case should be passed to either a human or model (but not both). The rejector thus can be seen as a meta-classifier that determines how to assign responsibility based on which decision maker (human or machine) is more likely to make the correct prediction. While L2D systems offer the promise of improved safety and robustness—by having a human available for support—this promise critically depends on the rejector's performance. Being a predictive model itself, the rejector is susceptible to the usual failure modes, such as distribution shift between training and test data.

In this paper, we perform principled uncertainty quantification for the rejector sub-component of L2D systems. Specifically, we use the framework of *conformal prediction* to allow the rejector to output sets, instead of just a single binary outcome (defer or not). This allows the rejector to express its uncertainty about whether the human or machine should be assigned to make the decision. In turn, this allows for safer decision making—for example, by abstaining from the prediction all together or querying *both* the human and model for their predictions. We report experimental results on tasks ranging from object to hate speech detection, showing that having an uncertain rejector can improve performance in uncertain cases via abstaining to make a prediction or checking for consensus between the human and model predictions.

## 2   BACKGROUND

### 2.1   Learning to Defer

**Setting, Data, and Model**   We focus on multiclass L2D (with one expert) [11, 14], though the ideas are presented can straightforwardly generalize to L2D-based regression [22]. Let $\mathcal{X}$ denote the feature space and $\mathcal{Y}$ the label space, a categorical encoding of $K \in \mathbb{N}^{\geq 2}$ classes. Let $\mathbf{x}_n \in \mathcal{X}$

Workshop on Bayesian Decision-making and Uncertainty, 38th Conference on Neural Information Processing Systems (NeurIPS 2024).

denote a feature vector, and $y_n \in \mathcal{Y}$ denotes the associated class index. L2D assumes that we have access to human predictions, denoted $m_n \in \mathcal{Y}$ for the associated feature vector $\mathbf{x}_n$. The training data then includes the features, the true label, and the human's prediction: $\mathcal{D} = \{\boldsymbol{x}_n, y_n, m_n\}_{n=1}^N$. The human is assumed to have some skill at the prediction task but is not an oracle. For example, the feature vector could be a medical image, $m_n$ is the expert's diagnosis from looking at the image, and $y_n$ is a true label that can only be obtained from a biopsy. L2D also assumes that the human has access to background knowledge that the classifier does not, such as years of medical training in the aforementioned example. The L2D framework requires two sub-models: a classifier and a rejector [7, 6]. We denote the *classifier* as $h : \mathcal{X} \to \mathcal{Y}$ and the *rejector* as $r : \mathcal{X} \to \{0, 1\}$. When $r(\mathbf{x}) = 0$, the classifier makes the decision, and when $r(\mathbf{x}) = 1$, the classifier abstains and defers the decision to the human. Thus the rejector can be thought of as a 'meta-classifier,' predicting which *predictor* would most likely be correct in its prediction.

**Learning** Learning in L2D requires we fit both the rejector and classifier. We assume that whoever makes the prediction—model or human—incurs a loss of zero (correct) or one (incorrect). Using the rejector to toggle between the human and model, we have the overall classifier-rejector loss:

$$L_{0-1}(h, r) = \mathbb{E}_{\mathbf{x},\mathbf{y},\mathbf{m}} \left[ (1 - r(\mathbf{x})) \, \mathbb{I}[h(\mathbf{x}) \neq \mathbf{y}] + r(\mathbf{x}) \, \mathbb{I}[\mathbf{m} \neq \mathbf{y}] \right] \tag{1}$$

where $\mathbb{I}[h(\mathbf{x}) \neq \mathbf{y}]$ denotes an indicator function that checks if the prediction and label are equal. Minimizing this loss results in the Bayes optimal classifier and rejector:

$$h^*(\boldsymbol{x}) = \arg\max_{y \in \mathcal{Y}} \mathbb{P}(\mathbf{y} = y | \boldsymbol{x}), r^*(\boldsymbol{x}) \quad = \mathbb{I} \left[ \mathbb{P}(\mathbf{m} = \mathbf{y} | \boldsymbol{x}) \geq \max_{y \in \mathcal{Y}} \mathbb{P}(\mathbf{y} = y | \boldsymbol{x}) \right] \tag{2}$$

where $\mathbb{P}(\mathbf{y} | \boldsymbol{x})$ is the probability of the label under the data generating process, and $\mathbb{P}(\mathbf{m} = \mathbf{y} | \boldsymbol{x})$ is the probability that the expert is correct. The assumption that the expert has additional knowledge is what allows it to possibly outperform the Bayes optimal classifier.

**Surrogate Losses** Several consistent surrogate losses have been proposed for Equation 1 [14, 20, 13, 12, 2, 4]. For our implementation, we focus on the two surrogates that have demonstrated the ability to learn calibrated predictors in practice—since the more calibrated the predictor, the better the conformal prediction results will be. Specifically, we use Verma and Nalisnick [20]'s one-vs-all (OvA) parameterization and Cao et al. [3]'s assymetric softmax (A-SM) parameterization. These parameterizations assume the classifier and rejector are unified via an augmented label space: $\mathcal{Y}^{\perp} = \mathcal{Y} \cup \{\perp\}$, where $\perp$ denotes the rejection option. Then let $g_k : \mathcal{X} \mapsto \mathbb{R}$ for $k \in [1, K]$ where $k$ denotes the class index, and let $g_{K+1} : \mathcal{X} \mapsto \mathbb{R}$ denote the rejection ($\perp$) option. The $g$ functions are analogous to the logits of a neural-network-based classifier. The OvA surrogate loss is given as [20]:

$$\psi_{\text{OvA}}(g_1, \dots, g_{K+1}; \boldsymbol{x}, y, m) = \phi[g_y(\boldsymbol{x})] + \sum_{y' \in \mathcal{Y}, y' \neq y} \phi[-g_{y'}(\boldsymbol{x})] + \phi[-g_{K+1}(\boldsymbol{x})]$$
$$+ \mathbb{I}[m = y] \left( \phi[g_{K+1}(\boldsymbol{x})] - \phi[-g_{K+1}(\boldsymbol{x})] \right) \tag{3}$$

where $\phi : \{\pm 1\} \times \mathbb{R} \mapsto \mathbb{R}_+$ is a binary surrogate loss. For instance, when $\phi$ is the logistic loss, we have $\phi[f(\boldsymbol{x})] = \log(1 + \exp\{-f(\boldsymbol{x})\})$. The A-SM surrogate loss is defined as follows [3]:

$$\psi_{\text{A-SM}}(g_1, \dots, g_{K+1}; \boldsymbol{x}, y, m) = -\log \phi_{\text{A-SM}}(g(\boldsymbol{x}), y) - \mathbb{I}[m \neq y] \cdot \log\left(1 - \phi_{\text{A-SM}}(g(\boldsymbol{x}), K+1)\right)$$
$$- \mathbb{I}[m = y] \cdot \log \phi_{\text{A-SM}}(g(\boldsymbol{x}), K+1) \tag{4}$$

where

$$\phi_{\text{A-SM}}(g(\boldsymbol{x}), y) = \begin{cases} \dfrac{\exp(g_y(\boldsymbol{x}))}{\sum_{y'=1}^{K} \exp(g_{y'}(\boldsymbol{x}))} & \text{if } y < K+1, \\ \dfrac{\exp(g_{K+1}(\boldsymbol{x}))}{\sum_{y'=1}^{K+1} \exp(g_{y'}(\boldsymbol{x})) - \max_{y' \in \mathcal{Y}} \exp(g_{y'}(\boldsymbol{x}))} & \text{otherwise.} \end{cases}$$

Here the 'asymmetry' is due to the softmax having different terms in the denominator for the class and rejector terms. The symmetric softmax parameterization [14] has the same denominator for both terms, which leads to issues for estimating the expert's correctness probability in practice [20, 3]. For both parameterizations, at test time, the classifier is obtained by taking the maximum over $g$ functions: $\hat{y} = h(\boldsymbol{x}) = \arg\max_{k \in [1,K]} g_k(\boldsymbol{x})$. The rejection function is given as: $r(\boldsymbol{x}) = \mathbb{I}[g_{K+1}(\boldsymbol{x}) \geq \max_k g_k(\boldsymbol{x})]$.

## 2.2 Conformal Prediction

*Conformal prediction* (CP) is a distribution-free approach to uncertainty quantification with finite-sample guarantees [18]. Given a test-time feature vector $\mathbf{x}_{N+1}$, CP seeks to construct a prediction set $C(\mathbf{x}_{N+1}; \tau) \subseteq \mathcal{Y}$ such that the true label $\mathbf{y}_{N+1}$ is included with probability $1 - \alpha$: $\mathbb{P}\left(\mathbf{y}_{N+1} \in C\left(\mathbf{x}_{N+1}; \tau\right)\right) \geq 1 - \alpha$, for $\alpha \in [0, 1]$. $\tau$ is a parameter that controls the set size, as will be described below. This statement is a *marginal* guarantee, meaning that it will hold, on average, over test samples but will not necessarily hold for any particular sample. CP's aforementioned guarantee is built off the crucial assumption that the test data is drawn exchangeably with a calibration set. To compute the parameter $\tau$ that controls the prediction sets, the *split*-CP (a.k.a. *inductive* CP) algorithm [16] is a popular choice due to its computational and sample efficiency [9] and resemblance to the traditional workflow of hyperparameter tuning. Split-CP requires $\tau$ be fit to a held-out validation set, which must be drawn exchangeably with the test set for the CP coverage guarantee to hold. Given an already trained classifier whose softmax outputs are denoted $\boldsymbol{f}(\mathbf{x}) = [f_1(\mathbf{x}), \dots, f_K(\mathbf{x})]$. CP then requires a score function be chosen that quantifies how well the model's prediction conforms to the true label. Using the softmax confidence associated with the true label is a reasonable choice: $s\left(\mathbf{x}, \mathbf{y}; \boldsymbol{f}\right) = 1 - f_\mathbf{y}(\mathbf{x})$, where $f_\mathbf{y}(\mathbf{x})$ is the softmax score for the true label. Others exist that incorporate all dimensions that have higher confidence than the true label [17]. Split-CP then proceeds by evaluating $s\left(\mathbf{x}, \mathbf{y}; \boldsymbol{f}\right)$ on all points in the held-out set and setting $\hat{\tau}$ to be the $(1 - \alpha)$ quantile (with a finite-sample correction) of the empirical distribution of scores. For a test time point $\mathbf{x}_{N+1}$, the prediction set is constructed as: $C(\mathbf{x}_{N+1}) = \{j | f_j(\mathbf{x}_{N+1}) > 1 - \hat{\tau}\}$, which represents the softmax dimensions that outscore the threshold $1 - \hat{\tau}$. CP is commonly evaluated by checking that the desired coverage is achieved in practice while also having efficient set sizes. The latter is crucial since the CP guarantee is trivially met by choosing $C(\mathbf{x}_{N+1}; \tau) = \mathcal{Y}$ for $(1 - \alpha)\%$ of cases.

## 3 UNCERTAIN DEFERRAL VIA CONFORMAL PREDICTION

We will now apply the CP framework to quantify the uncertainty in the rejector sub-component of an L2D system. Concretely, instead of just outputting 0 (model) or 1 (human), we want the CP-based rejector to output a set $C_r\left(\mathbf{x}; \tau\right)$, which is an element of the superset $\{\{0\}, \{1\}, \{0, 1\}\}$. $C_r\left(\mathbf{x}; \tau\right) = \{0, 1\}$ means that the rejector is unsure if the decision should be allocated to the human or model. Thus, instead of *prediction* sets, we call the uncertainty set of the rejector a *deferral set*. In Section 3.2, we will discuss how to incorporate these sets into downstream decision making.

**Ideal Construction**  Recalling the Bayes optimal decision rule for the rejector (Equation 2), it would be ideal if $C_r\left(\mathbf{x}; \tau\right)$ could satisfy the guarantee: $\mathbb{P}\left(r^*\left(\mathbf{x}_{N+1}\right) \in C_r\left(\mathbf{x}_{N+1}; \tau\right)\right) \geq 1 - \alpha$, which means that, marginally, the probability that the output of the Bayes optimal rejector is in the set is at least $1 - \alpha$. Constructing an adaptive set via validation statistics, unfortunately, requires we have access to $\mathbb{P}(\mathbf{m} = \mathbf{y} | \boldsymbol{x})$ to compute a non-conformity score. Moreover, if we did have access to $\mathbb{P}(\mathbf{m} = \mathbf{y} | \boldsymbol{x})$ (or a close approximation), then we could exactly quantify the uncertainty in deferral by direct use of $\mathbb{P}(\mathbf{m} = \mathbf{y} | \boldsymbol{x})$ and have no need for CP.

**Practical Construction**  We instead consider constructing the set to capture an alternative quantity: $\mathbb{I}\left[\mathbf{m}_{N+1} = \mathbf{y}_{N+1}\right]$, an indicator function representing if the human will make the correct prediction. Similarly, we wish to construct prediction sets such that this binary variable will have a coverage guarantee:

$$\mathbb{P}\left(\mathbb{I}\left[\mathbf{m}_{N+1} = \mathbf{y}_{N+1}\right] \in C_r\left(\mathbf{x}_{N+1}; \tau\right)\right) \geq 1 - \alpha. \tag{5}$$

This statement is not equivalent to the one above since the expert could be correct (i.e. $\mathbb{I}\left[\mathbf{m}_{N+1} = \mathbf{y}_{N+1}\right] = 1$) but $\mathbb{P}(\mathbf{y} | \mathbf{x})$ still be a better predictive model (i.e. $r^*(\mathbf{x}) = 0$). In other words, this formulation is considering the expert's performance in isolation of the classifier's. However, the semantics are retained since $C_r\left(\mathbf{x}_{N+1}; \tau\right) = \{0\}$ means that the expert will likely be wrong and so using the classifier is either a good decision or not an inferior one (if the model would also be wrong). Conversely, $C_r\left(\mathbf{x}_{N+1}; \tau\right) = \{1\}$ means that the expert will likely make the correction prediction. If $C_r\left(\mathbf{x}_{N+1}; \tau\right) = \{0, 1\}$, then the prediction set is unsure if the expert will be correct and still suggests uncertainty in the deferral decision. This relaxation, importantly, allows us to define a conformity statistic from which to compute a practical set, as we will discuss below.

## 3.1 Constructing Deferral Sets

We can construct deferral sets that follow the guarantee in Equation 5 by treating the deferral decision as a binary classification problem of whether the expert will make the correct prediction. Fortunately, both aforementioned L2D parameterizations directly model the probability that the expert will be correct. For the OvA parameterization, this probability is directly parameterized by the $(K+1)$th binary classifier: $\hat{p}(\mathrm{m}=\mathrm{y}|\mathbf{x}) = \phi\left[g_{K+1}(\boldsymbol{x})\right] = (1 + \exp\{-g_{K+1}(\boldsymbol{x})\})^{-1}$, with the logistic loss again being assumed. The A-SM similarly uses the deferral score, but here the parameterization requires evaluating all $K+1$ functions:

$$\hat{p}(\mathrm{m}=\mathrm{y}|\mathbf{x}) = \phi_{\text{A-SM}}(g(\boldsymbol{x}), K+1) = \frac{\exp(g_{K+1}(\boldsymbol{x}))}{\sum_{y'=1}^{K+1} \exp(g_{y'}(\boldsymbol{x})) - \max_{y'\in\mathcal{Y}} \exp(g_{y'}(\boldsymbol{x}))} \quad (6)$$

Both estimators have been shown to be competitively calibrated [3]. Given these estimators, we construct the usual non-conformity score for binary classification:

$$s\left(\mathbf{x}, \mathrm{y}, \mathrm{m}; \hat{p}\right) = \begin{cases} 1 - \hat{p}(\mathrm{m}=\mathrm{y}|\mathbf{x}) & \text{if } \mathrm{m}=\mathrm{y} \\ \hat{p}(\mathrm{m}=\mathrm{y}|\mathbf{x}) & \text{if } \mathrm{m}\neq\mathrm{y}. \end{cases} \quad (7)$$

To obtain the threshold $\hat{\tau}$, one would follow the standard procedure of computing these non-conformity scores on a validation set [1], obtaining the $(1-\alpha)$ empirical quantile, and then applying the threshold at test time as follows:

$$C_r\left(\mathbf{x}; \hat{\tau}\right) = \begin{cases} \{0\} & \text{if } 1 - \hat{p}(\mathrm{m}=\mathrm{y}|\mathbf{x}) \geq 1 - \hat{\tau} \\ \{1\} & \text{if } \hat{p}(\mathrm{m}=\mathrm{y}|\mathbf{x}) \geq 1 - \hat{\tau} \\ \{0,1\} & \text{otherwise} \end{cases} \quad (8)$$

## 3.2 Using Deferral Sets in Decision Making

Now that we have detailed how to construct CP deferral sets, we next address how to use them to improve decision making within the L2D framework. While there are surely alternative uses, below we detail three that we believe will be practical and useful in a variety of applications.

**Abstention** The use that likely first comes to mind is prediction with the option to abstain [5]. In the traditional case, the classifier only makes a prediction if it is confident; otherwise, it abstains. Our CP deferral sets allow for a similar workflow, but instead of abstaining because the prediction is uncertain, the L2D system will abstain because it is uncertain about to whom to allocate responsibility, the machine or human. Specifically, if $C_r\left(\boldsymbol{x}_{N+1}; \hat{\tau}\right) = \{0,1\}$, then the L2D system will abstain. Otherwise, the system will defer if $r^*(\boldsymbol{x}) = 1$.

**Consensus Prediction** We next consider how to make a prediction even if $C_r\left(\boldsymbol{x}_{N+1}; \hat{\tau}\right) = \{0,1\}$. If the rejector is uncertain to defer or not, we propose querying both the model and human for their predictions. If they agree, then that consensus prediction is output as the L2D system's final prediction. If they do not agree, then the system abstains from making any prediction. This workflow has the same appeal to safety as the abstention-only option, but it will likely have higher coverage since it will make predictions when the abstention-only workflow would not.

## 4 EXPERIMENTS

We now experimentally demonstrate that incorporating uncertainty into the deferral decision via CP can have tangible benefits to the safety and robustness of L2D systems. Our experiments follow closely the setup in previous works on L2D [15, 21, 3] for base models, experts simulation, data processing, training and hyperparameters, while introducing uncertainty quantification for the rejector. We trained L2D models using the OvA and A-SM surrogate losses. Taking this base L2D model, we then apply the CP procedure described in Section 3. We utilize three datasets tailored to different tasks: CIFAR-10 [10] for object detection, HAM10000 [19] for skin lesion diagnosis, and Hate Speech [8] for hate speech detection.

Table 1: *Performance Comparison.* The left table reports coverage and efficiency of conformal prediction given confidence level $1 - \alpha = 90\%$, while the right table focuses on abstention and consensus prediction metrics.

### (a) Coverage and Efficiency

| Dataset | Param. | Coverage (%) | Avg. Size |
|---|---|---|---|
| CIFAR-10 | OvA | $86.94 \pm 0.86$ | $1.07 \pm 0.03$ |
| | A-SM | $90.53 \pm 0.56$ | $1.37 \pm 0.01$ |
| HAM10k | OvA | $90.65 \pm 0.63$ | $1.25 \pm 0.01$ |
| | A-SM | $91.13 \pm 0.58$ | $1.28 \pm 0.03$ |
| HateSpeech | OvA | $90.35 \pm 0.53$ | $1.03 \pm 0.03$ |
| | A-SM | $90.67 \pm 0.52$ | $1.01 \pm 0.01$ |

### (b) Abstention and Consensus Prediction

| | Param. | Method | Sys. Acc. | Ratio Deferred | Sys. Cov. |
|---|---|---|---|---|---|
| CIFAR-10 | OvA | Base Model | $84.71 \pm 0.46$ | $55.26 \pm 1.76$ | 100 |
| | | Abstention | $86.72 \pm 1.02$ | $56.41 \pm 2.30$ | $92.14 \pm 0.48$ |
| | | Consensus | $\mathbf{86.79} \pm 1.07$ | $56.38 \pm 2.31$ | $93.32 \pm 0.52$ |
| | A-SM | Base Model | $84.01 \pm 0.45$ | $56.63 \pm 3.73$ | 100 |
| | | Abstention | $87.05 \pm 0.76$ | $84.13 \pm 4.56$ | $62.53 \pm 0.75$ |
| | | Consensus | $\mathbf{87.58} \pm 0.61$ | $79.62 \pm 4.31$ | $67.57 \pm 0.75$ |
| HAM10k | OvA | Base Model | $82.1 \pm 0.49$ | $33.71 \pm 2.39$ | 100 |
| | | Abstention | $\mathbf{87.48} \pm 0.51$ | $35.91 \pm 2.84$ | $75.23 \pm 1.40$ |
| | | Consensus | $85.72 \pm 0.63$ | $34.27 \pm 2.52$ | $88.39 \pm 1.85$ |
| | A-SM | Base Model | $78.92 \pm 0.29$ | $26.68 \pm 3.07$ | 100 |
| | | Abstention | $\mathbf{87.05} \pm 0.87$ | $28.11 \pm 3.45$ | $72.82 \pm 1.19$ |
| | | Consensus | $84.76 \pm 0.44$ | $27.49 \pm 3.16$ | $84.48 \pm 0.95$ |
| Hate Speech | OvA | Base Model | $92.09 \pm 0.07$ | $42.41 \pm 0.99$ | 100 |
| | | Abstention | $\mathbf{92.28} \pm 0.14$ | $42.48 \pm 0.96$ | $99.38 \pm 0.43$ |
| | | Consensus | $92.25 \pm 0.13$ | $42.42 \pm 0.96$ | $99.78 \pm 0.22$ |
| | A-SM | Base Model | $91.82 \pm 0.32$ | $67.91 \pm 1.76$ | 100 |
| | | Abstention | $\mathbf{91.88} \pm 0.15$ | $67.79 \pm 1.74$ | $99.16 \pm 0.75$ |
| | | Consensus | $\mathbf{91.88} \pm 0.12$ | $67.81 \pm 1.73$ | $99.65 \pm 0.28$ |

**Coverage and Efficiency** We experimentally verify that the target coverage is met, validating CP's guarantee (Equation 5). In Table 1, we report the empirical coverage and average set size for the three aforementioned datasets. Both parameterizations meet the target coverage level ($90\%$) for all datasets except for OvA on `CIFAR-10` ($\sim 87\%$). In all cases, the sets are quite efficient, with the average set size always being less than $1.3$. The exceptionally small set size of $1.07$ for OvA on `CIFAR-10` leads to its mis-coverage. We suspect the mis-coverage is due to (natural) train-test distribution shift.

**L2D with Abstention and Consensus** We next investigate the efficacy of the abstention and consensus decision making workflows presented in Section 3.2. In Table 1, we report the system accuracy, ratio of test points deferred, and the coverage of the system (i.e. the fraction of points for which the system does not abstain) again for `CIFAR-10`, `HateSpeech`, and `HAM10000`. We see that both OvA and A-SM improve upon the accuracy of the base L2D model for `CIFAR-10` and `HAM10000`, with improvements ranging from $2\%$ to $5\%$. However, the coverage reduction is variable, ranging from modest ($-8\%$) to substantial ($-38\%$), meaning that the accuracy improvement would be practical in some cases (e.g. OvA for `CIFAR-10`) and not in other (e.g. A-SM for `CIFAR-10`). On `Hate Speech`, very few of the points were abstained, leading to uninformative accuracy results. We do not see a clear superiority between the parameterizations.

## 5 Conclusions

In this paper, we have introduced an uncertainty-based method designed to enhance existing L2D systems and help their rejectors incorporate uncertainty. However, the conformal scoring function shall be carefully parameterized to best present the probability of the expert making the correct prediction. Future work could explore the extent to which this method helps maintain system safety and robustness across various failure modes, such as distribution shifts and shifts in human's predictions.

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
