# OpenReview forum: "Learning to Defer with an Uncertain Rejector via Conformal Prediction"
_NeurIPS.cc/2024/Workshop/BDU — NeurIPS BDU Workshop 2024 Poster_

### Official Review · Reviewer_Dt7d · 2024-09-14

**Rating:** 6
**Confidence:** 4

**Review:**

Summary: This paper uses conformal prediction for the rejector sub-component of the Learning-to-Defer (L2D) framework, allowing the rejector to output a set of possible outcomes rather than just a binary decision (defer or not). This method improves the safety and robustness of L2D systems by allowing the rejector to express its uncertainty.

Strengths:
1. The idea of combining CP with the L2D framework is unique and promising. Explicitly quantifying uncertainty allows the system to handle decision-making under uncertainty more robustly.
2. The problem setting is important and relevant to real-world scenarios, demonstrating how CP can be applied beyond simple uncertainty quantification to enhance decision-making in human-AI collaboration systems.
3. The paper is well-written and maintains a high level of technical soundness, presenting the concepts clearly and justifying the proposed approach.

Weaknesses:
1. While the idea is promising, the paper could benefit from a more in-depth discussion on different kinds of non-conformity scores, such as APS [1] and RAPS [2], as mentioned in the conclusion.
2. Integrating CP into the L2D framework may introduce additional computational complexity, which is not fully addressed or evaluated, especially in scenarios requiring real-time decision-making.

[1] Romano, Yaniv, Matteo Sesia, and Emmanuel Candes. "Classification with valid and adaptive coverage." Advances in Neural Information Processing Systems 33 (2020): 3581-3591.

[2] Angelopoulos, Anastasios Nikolas, et al. "Uncertainty Sets for Image Classifiers using Conformal Prediction." International Conference on Learning Representations.

---

### Official Review · Reviewer_MbJr · 2024-09-16

[review text omitted: it was posted to a different submission]

---

### Decision · Program_Chairs · 2024-10-09

Accept (Poster)